# Baseline Blood Levels of Mucin-1 Are Associated with Crucial On-Treatment Adverse Outcomes in Patients with Idiopathic Pulmonary Fibrosis Receiving Antifibrotic Pirfenidone

**DOI:** 10.3390/biomedicines12020402

**Published:** 2024-02-08

**Authors:** Tang-Hsiu Huang, Sheng-Huan Wei, Hung-I Kuo, Hsin-Yu Hou, Chin-Wei Kuo, Yau-Lin Tseng, Sheng-Hsiang Lin, Chao-Liang Wu

**Affiliations:** 1Institute of Clinical Medicine, College of Medicine, National Cheng Kung University, 35 Siaodong Rd., Tainan 704, Taiwan; tangomycin0713@gmail.com (T.-H.H.); kbh557@gmail.com (C.-W.K.); shlin922@ncku.edu.tw (S.-H.L.); 2Division of Chest Medicine, Department of Internal Medicine, National Cheng Kung University Hospital, College of Medicine, National Cheng Kung University, 138 Sheng Li Rd., Tainan 704, Taiwan; xtm0915@gmail.com (S.-H.W.); s5817627@yahoo.com.tw (H.-Y.H.); 3Chest Hospital, Ministry of Health and Welfare, 864 Zhongshan Rd., Rende Dist., Tainan 717, Taiwan; ahkuokuo@gmail.com; 4Division of Thoracic Surgery, Department of Surgery, National Cheng Kung University Hospital, College of Medicine, National Cheng Kung University, 138 Sheng Li Rd., Tainan 704, Taiwan; tsengyl@mail.ncku.edu.tw; 5Department of Public Health, College of Medicine, National Cheng Kung University, 138 Sheng Li Rd., Tainan 704, Taiwan; 6Biostatistics Consulting Center, National Cheng Kung University Hospital, College of Medicine, National Cheng Kung University, 138 Sheng Li Rd., Tainan 704, Taiwan; 7Department of Biochemistry and Molecular Biology, College of Medicine, National Cheng Kung University, 1 Dasyue Road, East District, Tainan 701, Taiwan; 8Ditmanson Medical Foundation Chia-Yi Christian Hospital, 539 Chung Hsiao Rd., Chiayi 600, Taiwan

**Keywords:** idiopathic pulmonary fibrosis, pirfenidone, mucin-1, acute exacerbation, mortality

## Abstract

Mucin-1 is a multi-functional glycoprotein expressed by type II alveolocytes and may be detectable in the circulation following pulmonary fibrosis. The prognostic utility of baseline pre-treatment blood levels of mucin-1 in patients with idiopathic pulmonary fibrosis (IPF) receiving antifibrotics has not yet been fully established. We retrospectively studied a cohort of patients (from two hospitals) with IPF who were receiving pirfenidone for >12 weeks. Baseline blood mucin-1 levels were measured via sandwich enzyme-linked immunosorbent assays. We investigated the performance of mucin-1 levels in longitudinally predicting the risks of acute exacerbation of IPF (AE-IPF) and severe adverse outcomes (SAO), including lung transplantation and death. Seventy patients were included; 20 developed AE-IPF; and 31 had SAO during the follow-up period. Patients with baseline mucin-1 levels ≥2.5 ng/mL had enhanced risks of AE-IPF (adjusted hazard ratio [aHR], 14.07; 95% confidence interval [CI], 4.26–46.49) and SAO within 2 years (aHR, 7.87; 95% CI, 2.86–21.70) and anytime during the follow-up (aHR, 4.68; 95% CI, 2.11–10.39). The risks increased across subgroups with increasing mucin-1 levels. Patients in the “mucin-1 ≥ 2.5” group also exhibited an accelerated decline in D_LCO_. This study supports baseline blood mucin-1 levels as a biomarker for IPF that predicts adverse outcomes during pirfenidone treatment.

## 1. Introduction

Idiopathic pulmonary fibrosis (IPF) is the archetypal fibrosing interstitial lung disease [1]. With its irreversibly progressive and destructive behavior, IPF causes significant impairment in quality of life and reduced survival [1,2,3]. The disease trajectory of IPF is also variable and may be complicated by life-threatening acute exacerbations (AE-IPF) [1,3,4,5]. The recent availability of two anti-fibrotic agents, pirfenidone and nintedanib, has marked a breakthrough in treating this difficult disease [1,3]. Pirfenidone slows down the progression of pulmonary fibrosis mainly by counteracting the stimulatory effects of transforming growth factor-beta on fibroblasts [1,6]. Clinical trials and real-world experience show that pirfenidone attenuates pulmonary functional decline [7,8,9,10,11,12,13,14,15,16,17]. Subsequent studies and meta-analyses have also found a potential beneficial effect of pirfenidone in lowering the risks of acute exacerbation [7,15,18], respiratory-related hospitalization [19], and on-treatment mortality [16,17,18,20,21,22,23]. However, like nintedanib, the therapeutic effects of pirfenidone are not universally observed among all patients [11,12,24,25]. There is as yet no widely accepted biomarker for predicting therapeutic responses during anti-fibrotic treatment.

Mucin-1 (alternatively known as Krebs von den Lungen-6 or KL-6) is a multi-functional transmembrane glycoprotein. In the lungs, mucin-1 is mainly expressed by type II alveolocytes [26,27] and is involved in pulmonary fibrogenesis [28,29]. The heavily glycosylated extracellular N-terminal subunit of mucin-1 may be shed and detectable in the circulation following pulmonary injury and fibrotic destruction of the alveolo–endothelial interface [27,30,31]. Mucin-1 (KL-6) has been proposed as a potential molecular biomarker for various interstitial lung diseases, including IPF [27,31,32,33,34,35,36,37]. For patients with IPF on antifibrotic treatments, prior studies have described the potential utility of serially measured blood levels of mucin-1/KL-6 in longitudinally assessing or predicting the therapeutic response [38,39,40,41,42]. We have previously reported that the single-time-point baseline (pre-treatment) plasma level of mucin-1/KL-6 is also prognostic and may serve as a clinical predictor of important unfavorable outcomes (specifically, drug-related hepatitis, on-treatment AE-IPF, and all-cause mortality) among patients with IPF receiving nintedanib [43]. It is unclear whether these findings regarding mucin-1 can be generalized to other populations and clinical scenarios, particularly to serve as an early prognostic alert for patients with IPF receiving different antifibrotic agents. Pirfenidone became available at National Cheng Kung University Hospital (NCKUH, a tertiary referral center in southern Taiwan) in June 2018, about 15 months after nintedanib. Since then, a distinct cohort of patients with IPF have been treated with this antifibrotic agent. In this study, we aimed to further explore the prognostic utility of the baseline pre-treatment measurement of mucin-1. We hypothesized that baseline blood levels of mucin-1 predict the risk of critical adverse outcome events (including AE-IPF, all-cause mortality, and lung transplantation) and accelerated pulmonary function decline in patients with IPF receiving pirfenidone treatment.

## 2. Methods

### 2.1. Design and Study Population

We tested our hypothesis by conducting this retrospective study involving patients with IPF who had been newly treated with pirfenidone and received regular follow-ups at NCKUH and at Chest Hospital (CH, a district hospital in southern Taiwan under the jurisdiction of the Ministry of Health and Welfare) between 1 June 2018, and 30 September 2023. The study was approved by the institutional review board of NCKUH (A-ER-107-193, A-ER-109-321, and A-ER-111-595). To be included in this study, patients had to be naïve to any antifibrotic therapy, be aged at least 50 years, have been diagnosed with IPF based on multi-disciplinary evaluations and international guidelines [2,44], and have received uninterrupted pirfenidone treatment for at least 12 weeks. We followed all included patients until death or 30 September 2023. Drug compliance was assessed during each outpatient visit and hospitalization. The electronic medical records for every patient were carefully reviewed and the following data were systematically retrieved: demographics; comorbidities (for the calculation of the Charlson comorbidity index [CCI]); smoking status; echocardiographic evidence of pulmonary hypertension; serial measurements of forced vital capacity (FVC) and diffusion capacity for carbon monoxide (D_LCO_) before and after pirfenidone treatment; gender-age-physiology (GAP) stages; pirfenidone dosing; and incidence and dates of AE-IPF, lung transplantation, and mortality.

### 2.2. Measurement of Blood Mucin-1 Levels

We followed our previously published protocols for blood specimen collection, specimen processing, and sandwich enzyme-linked immunosorbent assays (ELISA) for the measurement of blood mucin-1 (KL-6) levels [43]. Briefly, a blood specimen (8 mL) was collected from each patient (after they had provided formal informed consent) by specialized staff members from the Department of Pathology of both participating hospitals using an EDTA vacuum collection tube from Greiner Bio-One International, Kremsmünster, Austria. The specimen was immediately placed in an ice bath and rapidly transferred to a specialized refrigerator at NCKUH for storage at 4 °C. All subsequent processing of and experiments on the specimen were performed at the Core Research Laboratory at the Centre for Clinical Medical Research of NCKUH. Centrifugation (at 4 °C and 1000× *g* for 20 min, using an Eppendorf Centrifuge 5810R from Eppendorf SE, Hamburg, Germany) was performed within 4 h of collection. The supernatant plasma was carefully aspirated and divided among sterile Eppendorf tubes in a laminar flow bench under sterile conditions. The tubes were then immediately stored at −80 °C until use. The sandwich ELISA was performed using a specialized kit (product number EH0406) from Fine Test (Wuhan Fine Biotech, Wuhan, China), which deployed primary and secondary antibodies that specifically recognized the tandem-repeat domain of the extracellular N-subunit of mucin-1. The plasma was diluted two- or three-fold before measurement for each patient, and the experiment was repeated at least three times. The optical density absorbance at 450 nm was detected using a SpectraMax 340PC384 microplate reader from Molecular Devices, San Jose, CA, USA, operated using the SoftMax Pro software (version 5.4.1). An intra-assay coefficient of variation (CV) of <15% and an inter-assay CV of <15% were considered acceptable.

### 2.3. Important Definitions

The time interval between the date of initiation of pirfenidone therapy and either day 28 after the last dose of pirfenidone (for those who prematurely discontinued treatment) [9,43] or 30 September 2023 (for those who continued the treatment) was considered “on-treatment”. “High dose” and “low dose” referred to 1800 mg and 1200 mg of pirfenidone per day (in divided doses; these are the dosages approved by the Taiwan Food and Drug Administration), respectively [8]. “AE-IPF” was defined using previously published working definitions and specifically excluded events with identifiable infectious or non-infectious etiologies [5]. Patients who underwent lung transplantation or died from any cause while receiving pirfenidone (or within 28 days of the last dose) were defined as having had “severe adverse outcomes” (SAO). SAO that occurred within two years after pirfenidone treatment had begun were considered as “early SAO”. Pulmonary hypertension referred to an estimated systolic pulmonary arterial pressure (based on the tricuspid regurgitation jet velocity) of ≥35 mmHg, determined via transthoracic echocardiography [43,45]. When determining the annualized (52-week) and 24 week rates of change in FVC and D_LCO_, we used the formulae presented in Appendix A.

### 2.4. Statistical Analysis

We summarized categorical data as counts and percentages and continuous variables as means (standard deviation) if normally distributed, or medians (interquartile range [IQR]) if not normally distributed. We did not impute any missing data. Variables were compared between patient groups and tested for non-random differences using Fischer’s exact test or the Mann–Whitney U test, whichever was appropriate. Serial pulmonary function measurements were longitudinally assessed using the Wilcoxon signed-rank rest. The Shoenfeld test was used to test the assumption of proportional hazards. We conducted Kaplan–Meier analysis (with the log-rank test) and Cox proportional hazards regression analysis to assess the performance of candidate predictors in predicting the risks of adverse outcomes. For multi-variable regression analyses, in addition to the candidate predictor, we also incorporated the following theoretical potential confounders as covariables: GAP stage (as a quantified integration of age, sex, FVC, and D_LCO_); CCI; echocardiographic evidence of pulmonary hypertension; and smoking status. When analyzing the risk of AE-IPF, we controlled for the competing risk of on-treatment SAO by conducting the Fine–Gray subdistribution hazard regression. Sensitivity analyses were performed to assess the robustness of all the multi-variable models constructed. These analyses involved two parts. In the first part, the covariable “GAP stage” was replaced by its individual constituent variables (age, sex, FVC, and D_LCO_). In the second part, a hypothetical unidentified confounder of varying prevalence among the patient groups was introduced into all the multi-variable models. All statistical tests were two-tailed and *p* < 0.05 was considered statistically significant. Statistical analyses were performed using R (version 3.6.3) and SPSS (version 26). Graphs were drawn using MedCal (version 20.118).

## 3. Results

### 3.1. Study Population

Seventy-three patients (71 from NCKUH and two from CH) with IPF received pirfenidone treatment between 1 June 2018, and 30 September 2023. Seventy of these patients met the inclusion criteria and three (all from NCKUH) were excluded based on having a treatment duration and follow-up of < 12 weeks (Figure 1). The study cohort was male-predominant and had a mean age of 75.2 (± 9.5) years. Forty-five (64%) patients had a history of smoking and 25 (36%) had never smoked. The median duration of pirfenidone treatment was 55.9 weeks (IQR, 22.3–123.3). Thirty (43%) and 40 (57%) patients received the high- and low-dose regimens, respectively. The treatment was generally well tolerated. Reported adverse effects were mild and included pruritus (12 patients), decreased appetite (8 patients), non-postural dizziness (7 patients), nausea and vomiting (6 patients), and photosensitivity (5 patients). Unlike patients given nintedanib, drug-related hepatitis and diarrhea were not detected. Table 1 summarizes the baseline characteristics and outcomes of the study cohort.

### 3.2. Baseline Mucin-1 Levels and AE-IPF

Twenty (29%) patients developed on-treatment AE-IPF; the median time from treatment initiation to AE-IPF was 14.9 weeks (IQR, 7.9–50.6). Compared to those without AE-IPF, patients who developed AE-IPF had significantly higher blood levels of mucin-1 and lower D_LCO_ at baseline. They were also more likely to be classified as GAP stage 3 at baseline and to die during the follow-up (Figure 2a and Appendix A). The study cohort was then stratified by applying the cut-off value for blood mucin-1 levels (≥ or <2.5 ng/mL) that we have previously reported [43]. In the Kaplan–Meier analysis, patients with baseline blood mucin-1 levels ≥2.5 mg/mL had a significantly higher probability of experiencing AE-IPF during the follow-up (Figure 3a). The probability of developing AE-IPF during the follow-up also increased across patient groups with increasing baseline mucin-1 levels (Figure 3b). Cox proportional hazard regression analyses showed that these patients exhibited an increased risk of on-treatment AE-IPF after adjusting for potential confounders (the adjusted hazard ratio [aHR], 14.07; 95% confidence interval [CI], 4.26–46.49). When the study cohort was further stratified into subgroups with successively higher mucin-1 levels, we found that patients with higher baseline mucin-1 levels had greater hazard ratios of AE-IPF than patients with lower mucin-1 levels. These findings remained consistent after controlling for the competing risk of on-treatment SAO (Figure 3c and Appendix A).

### 3.3. Baseline Mucin-1 and SAO

Thirty-one (44%) patients developed on-treatment SAO during the follow-up, wherein 21 (30%; including the one patient who underwent lung transplantation) patients did within 2 years of pirfenidone treatment. The median time from pirfenidone initiation to SAO was 52.6 weeks (IQR, 15.0–112.6). About 80% of the on-treatment mortality was due to respiratory etiologies, and AE-IPF was the leading cause (Appendix A). Compared to those without SAO, patients who developed SAO had significantly higher blood levels of mucin-1 and lower D_LCO_ at baseline; an even higher baseline mucin-1 level was observed in patients who developed early SAO (Figure 2b,c, and Appendix A). The Kaplan–Meier analysis showed that patients with baseline plasma mucin-1 levels ≥ 2.5 ng/mL had a significantly lower survival probability (Figure 4a). Decreasing survival probability was also observed across patient groups with successively higher baseline mucin-1 levels (Figure 4b). After controlling for potential confounders, these patients had significantly enhanced risks of on-treatment early SAO (aHR 7.87; 95% CI, 2.86–21.70) and SAO anytime during the follow-up (aHR, 4.68; 95% CI, 2.11–10.39). These hazard ratios increased across subgroups with successively higher mucin-1 levels (Figure 4c,d, and Appendix A).

### 3.4. Sensitivity Analyses

Whether in the presence of a hypothetical unidentified confounder or after the covariable “GAP stage” had been replaced with its constituent variables (age, sex, FVC, and D_LCO_), all the sensitivity analyses yielded concordant results, which support the validity of the aforementioned findings (Appendix A).

### 3.5. Baseline Mucin-1 and Pulmonary Function Decline

Following the initiation of pirfenidone treatment, there was an overall pattern of reduction in the annualized decline in both FVC and D_LCO_ in the study cohort, although these differences did not reach statistical significance (Figure 5a,b). A comparison of the on-treatment annualized rates of change between the “mucin-1 ≥ 2.5” group and the “mucin-1 < 2.5” group revealed that, while there was no significant between-group difference in the annualized rate of change in FVC (either including or excluding patients with AE-IPF), the annualized rate of decline in D_LCO_ for the “mucin-1 ≥ 2.5” group was significantly greater than that of the “mucin-1 < 2.5” group, even after excluding patients with AE-IPF (Figure 5c–f). Furthermore, a numerically greater proportion of patients in the “mucin-1 ≥ 2.5” group exhibited a ≥ 10% decline in FVC over 24 weeks than in the “mucin-1 < 2.5” group. This trend, though, may be attributable to the functionally detrimental effect of on-treatment AE-IPF (Figure 5g). However, a significantly greater proportion of patients in the “mucin-1 ≥ 2.5” group suffered a 24-week decline of ≥ 10% in D_LCO_ than in the “mucin-1 < 2.5” group, and this finding persisted after patients with on-treatment AE-IPF had been excluded (Figure 5h).

## 4. Discussion

In the present study, which involved a real-world cohort of patients with IPF receiving antifibrotic pirfenidone, we found that baseline blood levels of mucin-1 were prognostic. Patients with baseline mucin-1 ≥ 2.5 ng/mL had enhanced risks of on-treatment AE-IPF and SAO (either within 2 years or anytime during the follow-up). These patients also exhibited a greater on-treatment decline in D_LCO_ than patients with mucin-1 < 2.5 ng/mL. These findings confirmed our hypothesis and are supportive of the prognostic utility of the single-time-point baseline check of mucin-1 for patients with IPF receiving antifibrotic treatments.

Currently, the diagnosis, severity assessment, and prognostic prediction for IPF rely mainly on pulmonary functional parameters and computed tomographic features. Until now, there has not been a widely accepted biochemical biomarker for IPF [1,2,3], although certain macromolecules in the blood—mucin-1/KL-6 in particular—have been proposed as potential candidates [27,31,35,36,37]. Pioneering studies before the “antifibrotic era” found that in patients with IPF, an elevated baseline blood level of mucin-1/KL-6 was associated with a greater decline in FVC [46], reduced survival [33,46], and an enhanced risk of AE-IPF [34]. Subsequently, Bergantini et al. analyzed serial measurements of blood mucin-1/KL-6 in nintedanib-treated patients with IPF and showed that mucin-1/KL-6 levels might become stable over time (up to 12 months) following antifibrotic treatment and that changes in serial mucin-1/KL-6 levels were correlated with variations in D_LCO_ [40]. d’Alessandro et al. followed a small cohort of nintedanib-treated patients for 24 months and concluded that patients exhibiting an incremental trend in serial mucin-1/KL-6 levels had an accelerated decline in their FVC [41]. For pirfenidone, although serial mucin-1/KL-6 levels were also measured in patients participating in the early pioneer clinical trials [7,8], the prognostic utility of this glycoprotein for therapeutic responses and outcomes was not further explored therein. Yoshikawa et al. [38] and Majewski et al. [39] separately studied different groups of patients with IPF who were receiving antifibrotic treatments (23 in the former study and 21 in the latter study received pirfenidone). Both studies revealed a negative correlation between serial measurements of mucin-1/KL-6 levels and pulmonary function parameters. The study by Yoshikawa et al. found that the median changes in mucin-1/KL-6 levels between the 3rd and the 6th month into pirfenidone treatment was significantly lower in the “stable-disease” group than in the “progressive-disease” group, whereas the study by Majewski et al. showed that an elevated baseline mucin-1/KL-6 level was associated with IPF progression. Recently, Choi et al. reported that the relative change in mucin-1/KL-6 levels at 1 month predicted disease progression (defined as relative decline in FVC ≥ 10% or DLco ≥ 15%, or AE-IPF, or deaths) over 6 months after antifibrotic treatment [42]. Compared to these previous findings, our present study is novel in demonstrating the prognostic utility of the single baseline measurement of blood mucin-1 levels. In our recent (involving nintedanib-treated patients with IPF, [43]) and present (involving a distinct cohort of pirfenidone-treated patients with IPF) studies, we showed that blood mucin-1 levels at baseline alone can already be prognostic. Our findings can potentially assist clinicians to anticipate the risks of crucial on-treatment adverse outcome events at the beginning of antifibrotic treatments, and thereby implement closer monitoring or other necessary measures in advance for those patients who are considered to have high risks based on their baseline mucin-1 levels.

Twenty (29%) patients in the present study developed AE-IPF during the follow-up period, which was relatively higher than the frequencies reported in many previous studies [1,7,8,12,13,14,16,19,47,48]. In our opinion, this difference may be attributable to the following factors. First, patients in our cohort, particularly those with on-treatment AE-IPF, had lower FVC and D_LCO_ (Table 1 and Appendix A) than those reported in many previous cohorts [7,8,12,13,14,16,19,47,48]. Poor pulmonary function is a known risk factor for AE-IPF [4,5,49]. Second, our patients were older than those in many previous cohorts [7,8,12,13,14,16,19,47,48] and tended to have multiple comorbidities (Table 1 and Appendix A). Advanced age may predispose patients to pulmonary infection. Finally, despite careful evaluation of the differential diagnosis, the distinction between AE-IPF and infection-triggered severe pulmonary inflammation is not always clear [5,49]. Therefore, some of the patients who were counted as having on-treatment AE-IPF might have actually had infectious pneumonia.

The present study has limitations. Owing to its retrospective design, this study may have been affected by unidentified confounders, despite the fact that substantial efforts were made to counteract this limitation (comprehensive data collection, strict statistical control of confounders, and sensitivity analyses of the major findings). Second, although ours was among the largest study cohorts ever assembled in Taiwan for IPF-related research, it was still relatively small. For a prospective study, it would be ideal to enroll a larger cohort of patients with more diverse demographic characteristics. However, that aim may make patient recruitment difficult, considering the epidemiological rarity of IPF. Third, a lower dose (1200 mg/day) was used for 40 (57%) of the patients due to their advanced age and side effects. This reflects real-world therapeutic practices. However, this appears not to have resulted in a confounding effect on our findings, as no significant difference in the proportions receiving each dosing was found across the patient groups (Appendix A). Further, comparable efficacy between high- and low-dose pirfenidone regimens has also been demonstrated [8,48].

## 5. Conclusions

In this study, we have shown that baseline blood levels of mucin-1 predicted the risk of an unfavorable prognosis in patients with IPF receiving pirfenidone treatment. Compared with patients with baseline mucin-1 < 2.5 ng/mL, those with mucin-1 ≥ 2.5 ng/mL had higher risks of on-treatment AE-IPF and SAO (either early or anytime during the follow-up). They also exhibited greater on-treatment deterioration in D_LCO_. Our findings contribute to evidence of the potential clinical utility of mucin-1 as a prognostic biomarker for IPF, particularly in identifying a susceptible subgroup for whom close monitoring for adverse outcomes and pulmonary function deterioration is critical, despite ongoing antifibrotic therapy.

## Figures and Tables

**Figure 1 biomedicines-12-00402-f001:**
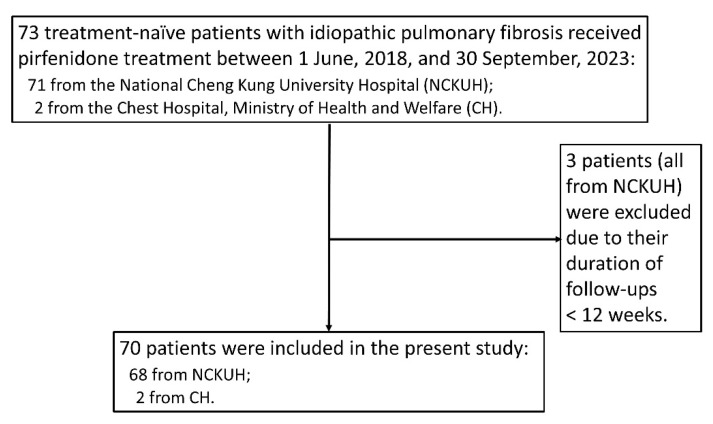
The flow chart of inclusion and exclusion of this study.

**Figure 2 biomedicines-12-00402-f002:**
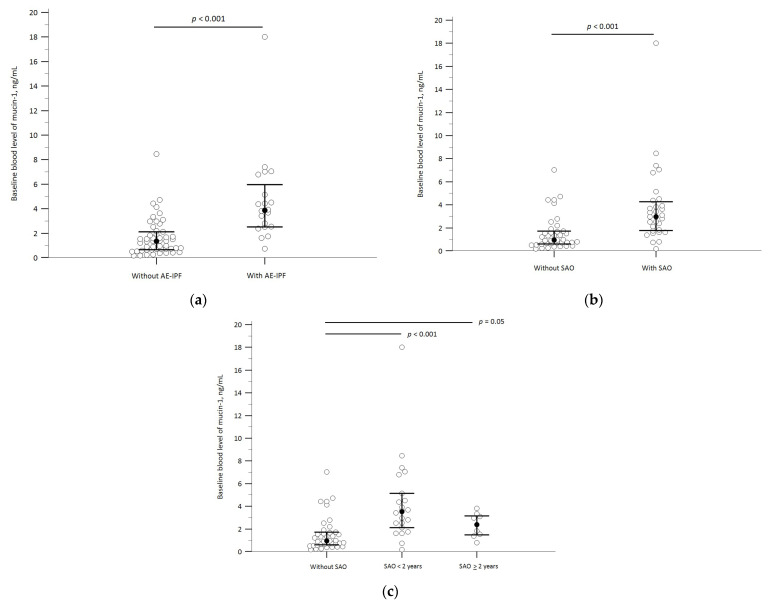
Baseline plasma mucin-1 levels of: (**a**) patients with AE-IPF versus those without AE-IPF; (**b**) patients with SAO versus those without SAO; and (**c**) patients without SAO versus those with SAO in <2 years or ≥2 years of pirfenidone treatment. The solid black dots and horizontal bars indicate medians and interquartile ranges, respectively. The Mann–Whitney U test was used for inter-group comparisons. Abbreviations: AE-IPF, acute exacerbation of idiopathic pulmonary fibrosis; SAO, severe adverse outcomes (including on-treatment lung transplantation and all-cause mortality).

**Figure 3 biomedicines-12-00402-f003:**
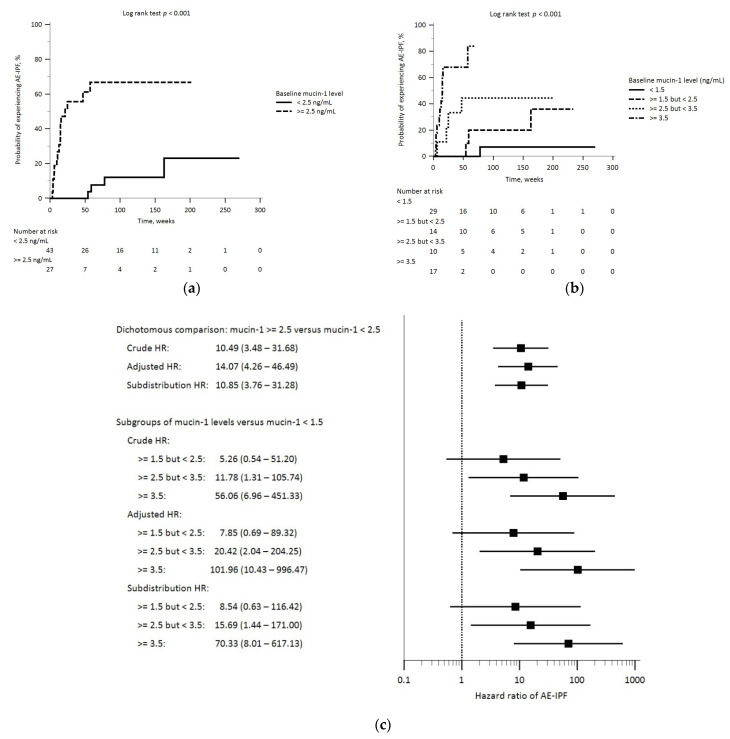
Legends: Kaplan–Meier analysis revealed a significant longitudinal difference in the probability of developing on-treatment AE-IPF: (**a**) between patient groups based on baseline plasma mucin-1 levels (≥2.5 versus <2.5, ng/mL); (**b**) among patient groups with successively increasing baseline levels of mucin-1; and (**c**) Cox proportional-hazard regression and subdistribution hazard regression analyses on the risk of on-treatment AE-IPF. Abbreviations: AE-IPF, acute exacerbation of idiopathic pulmonary fibrosis; HR, hazard ratio.

**Figure 4 biomedicines-12-00402-f004:**
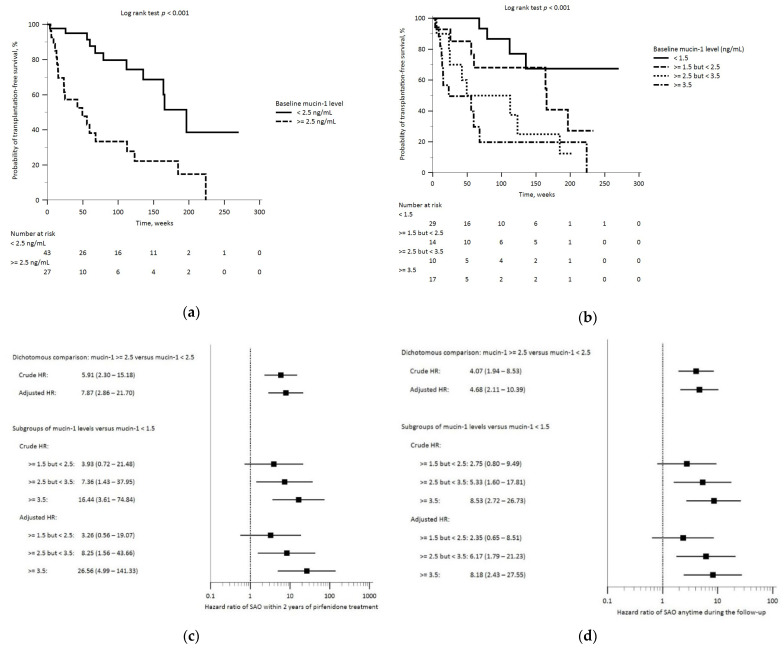
Legends: Kaplan–Meier analysis revealed a significant longitudinal difference in the probability of transplantation-free survival: (**a**) between patient groups based on baseline plasma mucin-1 levels (≥2.5 versus <2.5, ng/mL); and (**b**) among patient groups with successively increasing baseline levels of mucin-1. Cox proportional-hazard regression analyses on the risk of on-treatment: (**c**) early SAO (within 2 years after pirfenidone treatment had begun); and (**d**) SAO anytime throughout the follow-up period. Abbreviations: HR, hazard ratio; SAO, severe adverse outcomes (including on-treatment lung transplantation and all-cause mortality).

**Figure 5 biomedicines-12-00402-f005:**
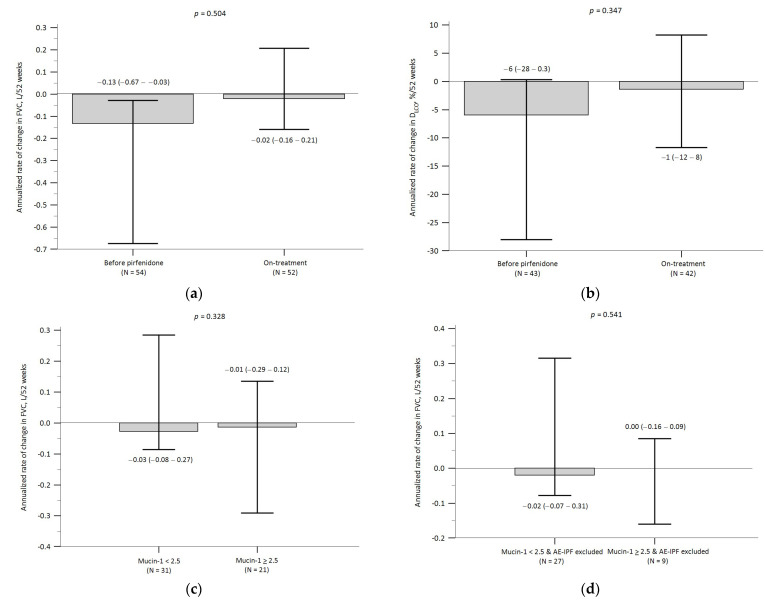
Legends: Comparison of the annualized rate of change in: (**a**) FVC of the whole cohort before and during pirfenidone treatment; (**b**) D_LCO_ of the whole cohort before and during pirfenidone treatment; (**c**) FVC during pirfenidone treatment between the “mucin-1 < 2.5” group and the “mucin-1 ≥ 2.5” group; (**d**) FVC during pirfenidone treatment between the “mucin-1 < 2.5” group and the “mucin-1 ≥ 2.5” group after excluding patients with on-treatment AE-IPF; (**e**) D_LCO_ during pirfenidone treatment between the “mucin-1 < 2.5” group and the “mucin-1 ≥ 2.5” group; and (**f**) D_LCO_ during pirfenidone treatment between the “mucin-1 < 2.5” group and the “mucin-1 ≥ 2.5” group after excluding patients with on-treatment AE-IPF. Comparison between the “mucin-1 < 2.5” group and the “mucin-1 ≥ 2.5” group (including and excluding patients with on-treatment AE-IPF) of the proportions of patients having ≥ 10% decline over 24 weeks in: (**g**) FVC; and (**h**) D_LCO_. Figure 5 Abbreviations: AE-IPF, acute exacerbation of idiopathic pulmonary fibrosis; D_LCO_, diffusion capacity of the lung for carbon monoxide; FVC, forced vital capacity.

**Table 1 biomedicines-12-00402-t001:** Baseline characteristics and on-treatment adverse outcomes of the 70 patients.

Baseline Characteristics and Outcome Events	Results
Age, years	75.2 (± 9.5)
Sex	
Female, *n* (%)	12 (17)
Male, *n* (%)	58 (83)
Body mass index, kg/m^2^	23.5 (±3.5)
Body surface area, m^2^	1.66 (±0.16)
Smoking status	
Never smoker, *n* (%)	25 (36)
Current smoker, *n* (%)	7 (10)
Former smoker, *n* (%)	38 (54)
Charlson comorbidity index	5 (4–6)
Echocardiographic evidence of pulmonary hypertension, *n* (%)	35 (50)
Baseline plasma mucin-1 level, ng/mL	1.64 (0.75–3.43)
Baseline FVC, L	1.98 (±0.50)
Baseline FVC, % predicted	66 (±11)
Baseline D_LCO_, mmol/min/kPa	2.72 (±1.23)
Baseline D_LCO_, % predicted	53 (±23)
Stages based on the GAP index	
Stage 1, *n* (%)	13 (19)
Stage 2, *n* (%)	45 (64)
Stage 3, *n* (%)	12 (17)
Dosing:	
Low (1200 mg/day), *n* (%)	40 (57)
High (1800 mg/day), *n* (%)	30 (43)
On-treatment AE-IPF, *n* (%)	20 (29)
On-treatment SAO:	
Anytime during the follow-up period, *n* (%)	31 (44)
All-cause mortality, *n* (%)	30 (43)
Lung transplantation, *n* (%)	1 (1)
Within 2 years following pirfenidone initiation, *n* (%)	22 (31)
All-cause mortality, *n* (%)	21 (30)
Lung transplantation, *n* (%)	1 (1)
Duration of pirfenidone therapy, weeks	55.9 (23.3–123.3)
Time to first on-treatment AE-IPF, weeks	14.9 (7.9–50.6)
Time to on-treatment mortality, weeks	52.6 (15.0–112.6)
Time to on-treatment lung transplantation, weeks	68.1

Categorical data are presented as counts and percentages, and continuous variables are presented as means (±standard deviation) or medians (interquartile range) if non-normally distributed. AE-IPF, acute exacerbation of idiopathic pulmonary fibrosis; D_LCO_, diffusion capacity for carbon monoxide; FVC, forced vital capacity; GAP, gender, age, physiology; SAO, severe adverse outcomes (including on-treatment lung transplantation and all-cause mortality).

## Data Availability

The de-identification datasets used and analyzed in the current study are available from the corresponding author upon reasonable request.

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
