# Peer review of "Baseline Blood Levels of Mucin-1 Are Associated with Crucial On-Treatment Adverse Outcomes in Patients with Idiopathic Pulmonary Fibrosis Receiving Antifibrotic Pirfenidone"

_biomedicines, 2024, doi:10.3390/biomedicines12020402_

Round 1

Reviewer 1 Report

Comments and Suggestions for Authors

To estimate the risk of AE-IPF, I recommend to reconsider the use of Kaplan Meirer analysis. Kaplan Meier are not designed to accommodate the competing nature of multiple causes and tend to produce inaccurate estimates when analyzing the marginal probability for cause-specific events. So, calculating the risk of AE-IPF using Kaplan Meirer may not be appropriate since patients who die are not at risk of AE-IPF. In this sense, it may be more appropriate to establish the calculation through the Cumulative Incidence Function (CIF) that estimates the probability of an event in the presence of competing events

Author Response

We thank the Reviewer for this important comment. We truly agree with the Reviewer about the impact of competing risks (which in our present study would be the risks of on-treatment severe adverse outcomes (SAO) including mortality and lung transplantation) on the incidence and risk of AE-IPF. We have also taken into consideration such impact of competing risks when planning the statistical methods for our present study. Therefore, when analyzing the risk of AE-IPF, in addition to single-variate Kaplan-Meier analysis and single-variate and multivariable Cox proportional hazard regression analyses, we have also performed Fine-Gray subdistribution hazard regression (using the R function “cmprsk”), which is an acceptable and widely used statistical method to control for the competing risks (Fine JP and Gray RJ. J Am Stat Assoc. 1999;94:496‐509). We have described this statistical step in Section 2.4 “Statistical analysis”, lines 13-15 of the original manuscript: “…When analyzing the risk of AE-IPF, we controlled for the competing risk of on-treatment SAO by conducting the Fine-Grey subdistribution hazard regression…”. Results of the Fine-Gray subdistribution hazard regression are presented as “subdistribution hazard ratios (subdistribution HR)” in the original Figure 3(c) and in the original Supplemental Tables S2 and S6, and described verbally in Section 3.2 “Baseline mucin-1 levels and AE-IPF”, lines 16-17 of the original manuscript: “…These findings remained consistent after controlling for the competing risk of on-treatment SAO…”.

In this revision, we have noticed that we had misspelled “Fine-Gray” as “Fine-Grey” in the original manuscript. We have corrected this misspelling in the revised manuscript (in Section 2.4 “Statistical analysis”, line 15 of the revised manuscript).

Reviewer 2 Report

Comments and Suggestions for Authors

The mucin-1 levels in 73 clinical blood samples from patients getting antifibrotic drug pirfenidone treatment were evaluated as a marker for pulmonary fibrosis to predict acute exacerbation. It is concluded that the risks increased across subgroups with increasing mucin-1 levels. The data supports the discussion and conclusions with appropriate statistical analysis. There is literature for the association between fibrosis and mucin-1 levels in blood. This manuscript presents new data on the association during treatment with an anti-fibrotic drug.

Author Response

We thank the Reviewer for this encouraging comment. We truly appreciate this opportunity to share with practitioners and researchers of clinical and basic medicine worldwide our research findings.  
